# Association between Organophosphate Pesticide Exposure and Insulin Resistance in Pesticide Sprayers and Nonfarmworkers

**DOI:** 10.3390/ijerph17218140

**Published:** 2020-11-04

**Authors:** Mathuramat Seesen, Roberto G. Lucchini, Somkiat Siriruttanapruk, Ratana Sapbamrer, Surat Hongsibsong, Susan Woskie, Pornpimol Kongtip

**Affiliations:** 1Department of Community Medicine, Faculty of Medicine, Chiang Mai University, Chiang Mai 50200, Thailand; lekratana56@gmail.com; 2School of Public Health, Florida International University, Miami, FL 33199, USA; rlucchin@fiu.edu; 3Occupational Medicine, University of Brescia, 25121 Brescia, Italy; 4Department of Disease Control, Ministry of Public Health, Nonthaburi 11000, Thailand; sirirut.somkiat@gmail.com; 5School of Health Science Research, Research Institute for Health Science, Chiang Mai University, Chiang Mai 50200, Thailand; surat.hongsibsong@cmu.ac.th; 6Department of Public Health, Zuckerberg College of Health Sciences, University of Massachusetts Lowell, Lowell, MA 01854, USA; Susan_Woskie@uml.edu; 7Department of Occupational Health and Safety, Faculty of Public Health, Mahidol University, Bangkok 10400, Thailand; pornpimol.kon@mahidol.ac.th; 8Center of Excellence on Environmental Health and Toxicology, Bangkok 10400, Thailand

**Keywords:** organophosphate, insulin resistance, dialkyl phosphate, pesticide sprayers, pesticide exposure, insecticides, metabolites

## Abstract

Insulin resistance is a risk factor for various diseases. Chronic organophosphate exposure has been reported to be a cause of insulin resistance in animal models. This cross-sectional study aimed to evaluate the association between organophosphate exposure and insulin resistance in pesticide sprayers and nonfarmworkers. Participants aged 40–60 years, consisting of 150 pesticide sprayers and 150 nonfarmworkers, were interviewed and assessed for their homeostatic model assessment of insulin resistance (HOMA-IR) level. Organophosphate (OP) exposure was measured in 37 sprayers and 46 nonfarmworkers by first morning urinary dialkyl phosphate (DAP) metabolites. The DAP metabolite levels were not different in either group except for diethylthiophosphate (DETP; *p* = 0.03), which was higher in sprayers. No significant association was observed between DAP metabolite levels and HOMA-IR. Wearing a mask while handling pesticides was associated with lower dimethyl metabolites (95% CI = −11.10, −0.17). Work practices of reading pesticide labels (95% CI = −81.47, −14.99) and washing hands after mixing pesticide (95% CI = −39.97, −3.35) correlated with lower diethyl alkylphosphate level. Overall, we did not observe any association between OP exposure and insulin resistance in pesticide sprayers and the general population. However, personal protective equipment (PPE) utilization and work practice were associated with OP exposure level in sprayers.

## 1. Introduction

Type 2 diabetes is one of the major global health problems and was reported by the World Health Organization (WHO) in 2018 as the seventh leading cause of death worldwide. Diabetes also leads to ischemic heart disease and stroke, which are reported as the first and second leading global cause of death, respectively [1]. In 2019, the world estimate for adult diabetic patients was 463 million, and the number is predicted to reach 700 million in 2045 [2]. In Thailand, the number of deaths from diabetes has increased dramatically during the past decade and reached 21.87 per 100,000 population in 2018 [3].

Insulin resistance is a significant underlying mechanism of metabolic syndrome [4] and type 2 diabetes [5]. Insulin resistance has also been reported to be a risk factor for cardiovascular disease [6,7] and cognitive impairment [8] and is associated with higher risk of endometrial [9], prostate [10], and colon cancer [11].

Genetic predisposition [12], obesity [13,14], and unhealthy lifestyle [15,16,17,18] represent other risk factors for insulin resistance. In addition, there is increasing evidence of chronic organophosphate (OP) exposure inducing insulin resistance in experimental animals [19,20,21,22].

As Thailand is an agricultural country and the world’s leading rice exporter, pesticides are widely used, especially in upland agriculture [23]. Thailand’s total pesticide use has reached the highest among the 10 Southeast Asian countries [24]. Most of the imported insecticides are OP chlorpyrifos. OP profenofos have also been reported among the top 10 imported insecticides in Thailand. In addition, the trend of OP imports in Thailand has increased steadily since 2011 [25].

Chiang Mai province in the northern region of Thailand is mostly covered by mountains and alluvial valleys. A study in 10 agricultural subdistricts in Chiang Mai found that 52% of soil samples were contaminated by pesticide, mostly chlorpyrifos. Moreover, OP metabolites were measured in the urine of farmworkers recruited from these areas [26]. At the same time, Chiang Mai province has the highest diabetic incidence in northern Thailand and is ranked fourth among the country’s provinces [27].

Studies on the effect of OP on human glucose level and insulin resistance are limited, and the available ones did not measure OP exposure through biomarkers [28,29,30]. To the best of our knowledge, there is only one human study of OP exposure and insulin resistance [31]. However, the target exposure of the study was malathion, which is different from the OPs commonly used in our area. In addition, other pesticide exposure and factors affecting insulin resistance were not measured. Therefore, this study aimed to determine the association between OP exposure and insulin resistance among pesticide sprayers and nonfarmworkers.

## 2. Materials and Methods

### 2.1. Study Design and Participants

A cross-sectional study was conducted in Chiang Mai province, Thailand, between May and July 2019, which is the period when the most intensive insecticide application occurs in agricultural villages. A community partnership that included village health volunteers, village headmen, and farmworker leaders was set up. People meeting the study criteria were recruited by the partnership. Participants aged 40–60 years and consisting of 150 pesticide sprayers from four agricultural villages and 150 subjects with no involvement in agricultural activity from three nonagricultural villages were enrolled. The inclusion criteria for the sprayers were having been pesticide sprayers for at least one year and having sprayed OP pesticides. The exclusion criteria were diagnosis of diabetes or consumption of antidiabetic medication, end-stage renal disease, cirrhosis, or consumption of HIV protease inhibitor. In addition, participants that were initially selected but were observed to have fasting plasma glucose of 126 mg/dL and higher were excluded.

Participants were requested to sign the consent form after full explanation of the study aims and methodology. After at least 8 h of fasting, each participant was interviewed and provided a blood sample to measure fasting blood glucose (FBG) and insulin levels.

A subset of 37 pesticide sprayers who had sprayed OPs in that season and planned to spray OPs within a month after blood collection as well as 46 nonfarmworkers were selected by convenience sampling for the measurement of urinary dialkyl phosphate (DAP) metabolites and lipid panel. To collect single morning urine sample, each participant was provided with a urine container and instructed for first morning urine collection in the following morning after OP spraying. Blood and urine collection as well as interviews were conducted for both farmworkers and nonfarmworkers at village temples.

Several subjects were excluded from the enrollment group. This included four farmers that were not pesticide sprayers, a sprayer with a small blood sample, and three sprayers and seven nonsprayers who had high FBG levels. The final group to be analyzed consisted of 142 sprayers and 143 nonsprayers, including 36 sprayers and 42 nonsprayers to be analyzed for urinary DAP metabolites.

### 2.2. Questionnaire

The questionnaire consisted of questions on sociodemographic and health information, 24 h dietary recall, physical activity, fresh produce consumption behavior, household pesticide use, work practice, and trade name of pesticides used (for pesticide sprayers).

A chart providing pictures of OPs and other pesticide containers that are generally used throughout Thailand was used for the interviews. Pesticide sprayers were asked to indicate the pesticide they have used in agricultural activities, while both groups of participants were asked to indicate household pesticide use.

Participants were asked to select the frequency of each activity regarding fresh produce consumption behavior. Pesticide sprayers were asked about frequency of use of personal protective equipment (PPE), which are gloves, boots, long-sleeved shirt, goggles, mask, and hat, and frequency of performing each activity while mixing or spraying pesticides.

Data from 24 h dietary recall was used to calculated daily calories by a nutritionist using the specialized Thai software NutriFact, which belongs to the Research Institute for Health Science, Chiang Mai University.

Information on duration and frequency of occupational activities, daily duration of walking or bicycling to work, and leisure-time activity was gathered. Participants who had done at least 150 min per week of moderate-intensity aerobic activity or 75 min per week of vigorous aerobic activity were classified as adequate physical activity.

The questionnaire was tested through pilot interview and refined by the research team and our community partnership. The local research team was trained by experts from the Research Institute for Health Science, Chiang Mai University. Questionnaires were administered through a face-to face interview by trained interviewers and research assistants from the Department of Community Medicine, Faculty of Medicine, Chiang Mai University.

### 2.3. Anthropometric Measurement

Waist circumference was measured at the midpoint between the lower margin of the lowest rib and the top of the iliac crest. Weight was measured with participants wearing light clothes and without shoes using a digital weighing scale. Height was measured without shoes using a stadiometer. The body mass index (BMI) was calculated as body weight (kg) divide by height squared (m^2^). Overweight and obesity was defined as having a BMI of 25 to < 30 and 30 and higher, respectively.

### 2.4. Exposure Measurement

First morning urine was collected and placed in zip-lock plastic bags, stored in cold boxes with dry ice, shipped to the laboratory at the Research Institute for Health Science, Chiang Mai University, and refrigerated at −20 °C until extraction within seven days and analyzed afterward.

The six urinary DAP metabolites of OP pesticides—dimethylphosphate (DMP), dimethylthiophosphate (DMTP), dimethyldithiophosphate (DMDTP), diethylphosphate (DEP), diethylthiophosphate (DETP), and diethyldithiophosphate (DEDTP)—were measured in urine samples using gas chromatography with a flame photometric detector (GC-FPD) according to the published method of Prapamontol et al. [32]. The measured DAP concentrations were adjusted by creatinine and converted from the unit μg/L to μg/g creatinine. Creatinine in urine was determined by Jaffe’s colorimetric reaction. Limit of detection (LOD) of DMP, DMPT, DMDPT, DEP, DETP, and DEDTP were 2.5, 0.4, 0.2, 0.25, 0.1, and 0.2 μg/L, respectively. Total DAP metabolites (∑DAPs) was calculated by summing the six individual DAP metabolites. Dimethyl alkylphosphate (DMAP) level was calculated by summing DMP, DMPT, and DMDPT. Diethyl alkylphosphate (DEAP) level was calculated by summing DEP, DETP, and DEDTP. To calculate ∑DAPs, DMAPs, and DEAPs, mass concentration of DAPs was converted to molar concentration. Undetectable DAP metabolites were substituted with LOD divided by the square root of 2 [33].

### 2.5. Outcome Measurement

Venipuncture was done after 8 h of fasting by nurses at village centers and transferred to laboratories at the Faculty of Associated Medical Sciences, Chiang Mai University, and the Maharaj Nakorn Chiang Mai Hospital for analysis. Abbott ARCHITECT c8000 analyzer was used to analyze plasma glucose, cholesterol, high-density lipoprotein (HDL) cholesterol, and triglyceride level by the hexokinase method, enzymatic method, accelerator selective detergent, and glycerol phosphate oxidase, respectively. Insulin level was evaluated by electrochemiluminescence immunoassay and Cobas e411. Insulin resistance was measured by homeostatic model assessment of insulin resistance (HOMA-IR), which was calculated with the following equation: HOMA-IR = (fasting glucose (mg/dL) × fasting insulin (μIU/mL))/405 [34]. Low-density lipoprotein (LDL) cholesterol was calculated using the Friedewald equation [35].

### 2.6. Statistical Analysis

Chi-square test for categorical variables and *t*-test for continuous variables were used to examine the demographic characteristics of participants. DAP metabolite levels in both pesticide sprayers and nonfarmworkers were analyzed using the Mann–Whitney *U* test. Agricultural product consumption behavior between pesticide sprayers and nonfarmworkers were compared using chi-square test. Multiple linear regression analysis was performed to examine the association between DAP metabolites and blood chemistry levels as well as DAP metabolites and agricultural work practice in farmworkers. *p*-value < 0.05 was considered statistically significant.

### 2.7. Ethical Consideration

All participants gave their informed consent for inclusion before participating in the study. The research procedure was approved by the Research Ethics Committee, Faculty of Medicine, Chiang Mai University, Thailand (Certificate Ethical Clearance No. 090/2019, 29 April 2019)

## 3. Results

### 3.1. Demographic Information, HOMA-IR, and Glucose Level of Pesticide Sprayers and Nonfarmworkers

The characteristics of participants by occupational status are presented in Table 1. Overall, pesticide sprayers were significantly older (*p* = 0.03), had a higher proportion of male (*p* = 0.02), and had lower education (*p* < 0.01) compared to nonfarmworkers. In addition, the sprayer group showed higher cardiometabolic risk factors, which were family history of diabetes (*p* < 0.01), smoking (*p* = 0.03), and excessive calorie (*p* < 0.01) and carbohydrate (*p* < 0.01) intake. Moreover, they reported higher hours of organophosphate pesticide exposure (*p* < 0.01), while no nonfarmworker reported OP exposure.

### 3.2. Assessment of DAP Metabolite Levels

Table 2 illustrates urinary DAP metabolites in the two groups. There was no significant difference in the proportion of participants who had DAP metabolite levels above and below LOD. In addition, DAP levels of both groups were similar except for DETP, which was higher in pesticide sprayers (*p* = 0.03).

### 3.3. Assessment of Agricultural Product Consumption Behavior

Assessment of agricultural product consumption behavior is presented in Table 3. A higher proportion of pesticide sprayers reported always consuming organic vegetables (*p* < 0.01), fruits (*p* < 0.01), and rice (*p* < 0.01). Moreover, a greater percentage of sprayers reported that they always have food cooked by themselves or their family members.

### 3.4. Factors Associated with HOMA-IR, Glucose, and Lipid Levels

Factors associated with HOMA-IR, fasting plasma glucose, and lipid levels in sprayers and the general population are shown in Table 4 and Table 5. HOMAR-IR had significant correlation with only waist circumference (95% CI = 0.03, 0.09). Participants who reported to have family history of diabetes showed lower glucose level (95% CI = −13.67, −1.87). For lipid panel, high amount of alcohol consumption correlated with high triglyceride (95% CI = 0.03, 0.12). Triglyceride level was also higher in subjects who had excessive calorie intake (95% CI = 30.84, 183.53). Furthermore, female tended to have higher HDL (95% CI = 2.86, 11.77). However, no significant correlation was observed between DAP and HOMAR-IR, glucose, triglyceride, LDL, and HDL cholesterol.

### 3.5. Factors Associated with DAP Metabolites in Pesticide Sprayers

Table 6 demonstrates factors associated with DAP metabolites in pesticide sprayers. For PPE utilization, gloves and boots were not analyzed as all sprayers reported to usually wear them. The association between higher frequency of wearing a mask while handling pesticide and DMAP level was observed (95% CI = −11.10, −0.17). Participants who reported lower frequency of reading pesticide labels (95% CI = −81.47, −14.99) and washing hands after mixing (95% CI = −39.97, −3.35) exhibited significant higher DEAP level.

## 4. Discussion

To the best of our knowledge, this is one of the first studies to determine the association between organophosphate exposure and insulin resistance in agricultural and nonagricultural workers. However, we could not demonstrate the correlation between organophosphate exposure and insulin resistance, fasting plasma glucose, triglyceride, HDL, or LDL cholesterol levels.

DAP metabolite levels of pesticide sprayers in this study were lower than previous studies, which could be from several possible causes. First, as different techniques were used to analyze DAP metabolites, LOD of our metabolites was higher than other studies. This was especially the case for DMP, which was 2.5 μg/L in our study compared to 0.26 μg/L in a study in the Netherlands [36]. This also led to the low percentage of DMP detection in our study. Another explanation could be that some farmworkers might not have followed the urine collecting instruction because of communication barriers as half of them were from the hill tribes.

DAP metabolites in the current study were also lower than that reported in a study of pesticide sprayers in Greece [37], which could be due to the lower proportion of PPE use. In the study, the percentage of sprayers who reported wearing gloves, hat, and mask was 35.5, 52, and 50%, respectively, whereas it was 94.4, 94.4, and 88.9%, respectively, in our study (data not shown). Similar to our result, the study pointed out that DETP level in sprayers was significantly higher than nonfarmworkers.

The average diethyl metabolite levels in our study were close to those in chili farmers in Thailand [38], where first morning urine was collected after spraying OPs. This could be the result of a similar DAP analyzing technique. Moreover, chlorpyrifos was commonly used among chili farmers in the study site.

Our study indicated that pesticide sprayers had significantly higher DETP metabolites than nonfarmworkers. This result is related to chlorpyrifos and ethion, the two most popular OPs used by pesticide sprayers, which can metabolize to DETP [39]. However, the levels of the other five metabolites were similar in both groups. This could be explained by the fact that nonfarmworkers had higher dietary OP pesticide exposure compared to pesticide sprayers as sprayers reported significantly higher organic produce consumption. This result correlates with several studies on the general population that have reported an association between DAP metabolites and higher fruit and vegetable intake [36,40,41]. Consumption of fresh produce contaminated with OP residues could be pointed as the main route of OP exposure in the general population, while organic diet consumption could reduce OP metabolites [42,43]. In addition, a study in Chiang Mai, Thailand, reported that the most common OP residue found in fresh produce was chlorpyrifos, which was detected in all kinds of fruit and vegetable samples [44].

Our study did not show an association between DAP metabolites and insulin resistance. This is consistent with a study on the US general population, which reported the lack of association between urinary DAP metabolites and HOMA-IR and plasma glucose [45]. This study result is in contrast with a previous research on pesticide applicators, which indicated the positive correlation between blood malathion concentration and HOMA-IR [32]. This could be explained by the questionable conclusion of this study that the waist circumference and BMI, which are mandatory factors of insulin resistance [46,47], of the farmer group were significantly higher than the control group. In addition, an association between lipid levels and OP exposure was not observed, whereas several studies in animal models have reported that OPs lead to an increase in LDL and triglyceride and a decrease in HDL [48,49]. Moreover, a study on humans showed the correlation between DMPT metabolite and higher HDL, lower LDL, and lower total cholesterol [45]. Thus, further human studies to examine the effect of OPs on lipid metabolism are needed.

Our study showed the inverse correlation between family history of diabetes and fasting plasma glucose level, which contradicts a previous study on the general population [50]. According to the Thai National Health Examination Survey, the proportion of undiagnosed diabetes was 51.2% for men and 41.3% for women [51]. Furthermore, some farmworkers in our study were of the hill tribe ethnic group, in which barriers to accessing healthcare services has been reported [52]. A survey on elderly Thai population pointed out that hill tribes had higher prevalence of diabetes compared to the general Thai population [53]. Therefore, the number of participants who reported family history of diabetes in our study might be lower than reality, especially in farmworkers. Another possible explanation could be greater awareness and risk perception in individuals who have familial risk of diabetes [54]; consequently, they are more likely to adopt healthy behaviors [55].

A reason behind the lack of association between OP exposure and chronic outcomes, namely, insulin resistance, glucose, and lipid levels, in our study could be the limitation of DAP metabolites. Although first morning urine has been pointed out as the best predictor of daily DAP metabolite concentration [56], various studies have reported that single urine analysis may not indicate chronic exposure [57,58]. Another reason could be the high exposure in the general population as over 90% of our nonfarmworkers had detectable DAP metabolites. Compared to studies among the general population in Israel and the US, our study showed higher diethyl alkyl metabolite levels [40,59]. Our participant enrollment process, in which the frequency of OP exposure in sprayers was not specified, could have affected the results. The frequency of pesticide spraying has been identified as being correlated with adverse health effects [60]. However, we analyzed the association between cumulative OP exposure and insulin resistance, but an association was not observed. In addition, there is evidence that exposure to carbamate pesticides alters glucose homeostasis in animals [61]. The current study did not assess carbamate levels because the sprayers reported that the most recent carbamate pesticide spraying was longer than a month before enrollment. Therefore, the metabolites could not be captured. This could probably be a cause for our negative correlation.

Our study reported that wearing a mask while spraying pesticides was associated with lower dimethyl metabolites. This result is consistent with several studies that have indicated a correlation between PPE utilization and lower DAP levels [37,62]. In addition, a positive correlation was found between diethyl metabolite level and the behavior of pesticide sprayers, such as reading pesticide labels before mixing and washing hands after mixing pesticides. This finding corresponds with a previous study that pointed out the association between the working practices of farmers and pesticide exposure levels [63].

There are several limitations of this study. Due to the complicated data collection process, participants were not randomly selected. Therefore, healthy people tended to be recruited because of awareness about their health. Our sample size was smaller than expected because of the difficulty of contacting farmworkers who lived in mountain villages. Another point to note is that the ability to represent chronic OP exposure and variation of urinary DAP metabolites over time might have been limited due to single morning urine collection. As our DAP analytical method had lower sensitivity compared to other studies, detectable metabolites in our study might not represent the real values. In addition, farmworkers might overreport PPE utilization and appropriate work practices as they were not observed while working.

## 5. Conclusions

This study did not observe an association between OP pesticide exposure and insulin resistance in pesticide sprayers and nonfarmworkers. However, the results are inconclusive due to limitation in detecting chronic exposure from single urinary DAP metabolites. In addition, the general population had similar OP pesticide exposure level, which was associated with consumption of fruit and vegetable contaminated with OP. The results therefore need further longitudinal studies with adequate sample size and multiple urine samples. Although there was no evidence of a correlation between measurable DAP metabolites and adverse health effect, the DAP levels reflect high exposure in our nonfarmworker group. Measures to control pesticide exposure should therefore be taken.

## Figures and Tables

**Table 1 ijerph-17-08140-t001:** Characteristics of participants according to occupational status.

Variable	Pesticide Sprayers (n = 142)	Nonfarmworkers (n = 143)	*p*-Value
Age (mean ± SD) ^a^	50.15 ± 6.35	51.71 ± 5.80	0.03 *
Sex, *n* (%) ^b^			0.02 *
Male	57 (40.1)	39 (27.3)	
Female	85 (66.3)	104 (72.7)	
BMI, *n* (%) ^b^			0.32
Normal	52 (36.6)	44 (30.8)	
Overweight	33 (23.2)	29 (20.3)	
Obese	57 (40.1)	70 (49)	
Educational level, *n* (%) ^b^			<0.01 *
Less than primary school	68 (47.9)	30 (21.0)	
Primary school	34 (23.9)	49 (34.3)	
High school	33 (23.2)	45 (31.5)	
High vocational and college	7 (4.9)	19 (13.3)	
Family history of diabetes, *n* (%) ^b^	107 (75.4)	85 (59.4)	<0.01 *
Excessive alcohol consumption, *n* (%) ^b^	32 (22.5)	27 (18.9%)	0.44
Current smoking, *n* (%) ^b^	25 (38)	13 (9.1)	0.03 *
Adequate physical activity, *n* (%) ^b^	127 (89.4)	120 (83.9%)	0.17
Excessive calorie intake, *n* (%) ^b^	37 (26.1)	19 (13.3)	<0.01 *
Excessive carbohydrate intake, *n* (%) ^b^	79 (55.6)	38 (26.6)	<0.01 *
Cumulative OP exposure (hours) (mean ± SD) ^a^	447.06 ± 2919.69	0 ± 0	<0.01 *
HOMA-IR (mean ± SD) ^a^	1.48 ± 1.27	2.30 ± 2.88	<0.01 *
Abnormal HOMA-IR, *n* (%) ^b^	32 (22.5)	54 (37.8)	<0.01 *
Fasting blood glucose (mean ± SD) ^a^	86.17 ± 12.21	90.88 ± 11.84	<0.01 *
Abnormal fasting blood glucose, *n* (%) ^b^	17 (12)	25 (17.5)	0.18

^a^ tested as *t*-test, ^b^ tested as chi-square. OP: organophosphate; HOMA-IR: homeostatic model assessment of insulin resistance, * *p* < 0.05.

**Table 2 ijerph-17-08140-t002:** DAP levels (μg/g creatinine) and detection frequency in pesticide sprayers and nonfarmworkers.

Metabolites (μg/g creatinine)	Pesticide Sprayers (n = 36)		Nonfarmworkers (n = 42)		*p*-Value
	Median (Range)	Detection Frequency, *n* (%)	Median (Range)	Detection Frequency, *n* (%)	
DMP	1.59 (0.57, 17.18)	1 (2.8)	1.96 (0.65,15.94)	2 (4.8)	0.07
DMTP	0.30 (0.10, 3.92)	5 (13.9)	0.36 (0.10, 5.72)	7 (16.7)	0.25
DMDTP	0.13 (0.05, 0.77)	1 (2.8)	0.15 (0.05, 0.56)	0 (0)	0.20
DEP	1.73 (0.06, 25.81)	28 (77.8)	1.68 (0.30, 11.58)	39 (92.9)	0.77
DETP	1.29 (0.03, 32.73)	31 (86.1)	0.61 (0.06, 10.94)	35 (83.3)	0.03 *
DEDTP	0.14 (0.05, 18.12)	9 (25)	0.16 (0.05, 1.65)	2 (4.8)	0.99
∑DAP	7.30 (1.19, 74.37)	34 (94.4)	6.63 (2.81, 18.02)	40 (95.2)	0.69

Dialkyl phosphate (DAP) metabolite levels were analyzed using the Mann–Whitney *U* test. DMP: dimethylphosphate; DMTP: dimethylthiophosphate; DMDTP: dimethyldithiophosphate; DEP: diethylphosphate; DETP: diethylthiophosphate; DEDTP: diethyldithiophosphate; ∑DAPs: total DAP metabolites, * *p* < 0.05.

**Table 3 ijerph-17-08140-t003:** Agricultural consumption behavior in pesticide sprayers and nonfarmworkers.

Behavior	Pesticide Sprayers, *n* (%)	Nonfarmworkers, *n* (%)	*p*-Value
Organic vegetable consumption			<0.01 *
Always	63 (44.4)	35 (24.5)	
Sometimes	43 (30.3)	44 (30.8)	
Rarely	36 (25.4)	64 (44.8)	
Organic fruit consumption			<0.01 *
Always	40 (28.2)	22 (15.4)	
Sometimes	57 (40.1)	42 (29.4)	
Rarely	45 (31.7)	79 (55.2)	
Organic rice consumption			<0.01 *
Always	38 (26.8)	20 (14.0)	
Sometimes	34 (23.9)	26 (18.2)	
Rarely	70 (49.3)	97 (67.8)	
Eat food cooked by themselves or their family members			<0.01 *
Always	134 (94.4)	106 (74.1)	
Sometimes	6 (4.2)	31 (21.7)	
Rarely	2 (1.4)	6 (4.2)	
Wash vegetables and fruits before eating			0.19
Always	136 (95.8)	134 (93.7)	
Sometimes	3 (2.1)	1 (0.7)	
Rarely	3 (2.1)	8 (5.6)	

* *p* < 0.05.

**Table 4 ijerph-17-08140-t004:** Factors associated with HOMA-IR and fasting blood glucose in sprayers and the general population.

	HOMA-IR			Fasting Blood Glucose		
	Beta	SE	95% CI	Beta	SE	95% CI
Age	0.07	0.02	−0.03, 0.05	0.19	0.23	−0.10, 0.84
Sex	−0.06	0.32	−0.08, 0.47	−0.22	3.45	−12.52, 1.28
Educational level	0.10	0.14	−0.14, 0.42	0.15	1.50	−1.14, 4.85
Family history of diabetes	−0.05	0.28	−0.69, 0.42	−0.31	2.95	−13.67, −1.87 *
Excessive calorie intake	0.09	0.33	−0.37, 0.98	0.06	3.58	−5.27, 9.04
Physical activity	0.06	0.52	−0.75, 1.34	−0.10	5.55	−15.51, 6.68
Smoking status	0.004	0.60	−1.18, 1.22	−0.09	6.38	−17.21, 8.27
Diuretic taking duration	0.06	0.44	−0.65, 1.12	−0.04	4.70	−10.98, 7.81
Alcohol consumption	−0.14	0.00	−0.001, 0.00	−0.05	0.002	−0.005, 0.003
Waist circumference	0.46	0.01	0.03, 0.09 *	0.11	0.15	−0.16, 0.46
∑DAP	−0.20	0.01	−0.05, 0.001	0.06	0.14	−0.20, 0.37

* Statistical significance.

**Table 5 ijerph-17-08140-t005:** Factors associated with lipid levels in sprayers and the general population.

	LDL			Triglyceride			HDL		
	Beta	SE	95% CI	Beta	SE	95% CI	Beta	SE	95% CI
Age	0.13	0.73	−0.65, 2.27	0.04	2.53	−4.04, 6.08	−0.06	0.17	−0.44, 0.23
Sex	0.22	9.53	−1.38, 36.70	−0.11	32.98	−98.50, 33.15	0.36	2.23	2.86, 11.77 *
Educational level	0.23	4.61	−0.58, 17.84	−0.004	15.99	−32.56, 31.28	−0.02	1.08	−2.42, 1.89
Excessive calorie intake	−0.01	11.02	−23.13, 20.89	0.30	38.24	30.84, 183.53 *	−0.32	2.58	−12.95, −2.62
Physical activity	−0.07	14.83	−38.90, 20.34	0.19	51.51	−9.52, 196.14	0.13	3.48	−2.69, 11.22
Lipid-lowering medication	−0.17	13.54	−46.36, 8.74	0.09	46.97	−52.99, 134.53	−0.12	3.17	−9.83, 2.85
Alcohol consumption	−0.23	0.006	−0.02, 0.00	0.37	0.02	0.03, 0.12 *	−0.21	0.001	−0.006, 0.00
∑DAP	0.04	0.40	−0.65, 0.96	−0.94	1.40	−4.00, 1.60	0.12	0.09	−0.08, 0.29

* Statistical significance.

**Table 6 ijerph-17-08140-t006:** Factors associated with DAP metabolites in pesticide sprayers.

Variables	DAPs			DMAPs			DEAPs		
	Beta	SE	95% CI	Beta	SE	95% CI	Beta	SE	95% CI
Age	−0.18	0.57	−1.61, 0.96	0.04	0.18	−0.38, 0.43	−0.19	0.48	−1.43, 0.73
Sex	0.04	6.00	−12.38, 14.39	0.57	1.90	−0.26, 8.21	−0.12	5.04	−14.21, 8.26
Education level	−0.15	2.88	−8.22, 4.61	−0.17	0.91	−2.63, 1.43	−0.10	2.41	−6.59, 4.18
Smoking status	0.18	11.91	−19.26, 33.81	0.02	43.77	−8.10, 8.70	0.18	10.00	−15.31, 29.26
Duration of spraying pesticides	−0.30	0.46	−1.44, 0.60	−0.41	0.14	−0.49, 0.15	−0.17	0.38	−1.11, 6.12
Long-sleeved shirt	0.18	16.91	−25.73, 49.63	−0.38	5.35	−19.26, 4.60	0.30	14.20	−12.36, 50.93
Glasses	−0.20	6.92	−20.16, 10.69	−0.11	2.19	−5.65, 4.12	−0.17	5.81	−16.93, 8.99
Mask	−0.05	7.74	−19.26, 15.24	−0.56	2.45	−11.10, −0.17 *	0.10	6.50	−10.86, 18.11
Hat	0.19	18.41	−28.26, 53.75	−0.29	5.83	−18.56, 7.42	0.28	15.46	−16.12, 52.78
Eat during spraying	−0.03	12.29	−28.73, 26.06	−0.26	3.89	−11.64, 5.71	0.04	10.33	−21.38, 24.65
Wash hands before eating	0.25	8.81	−13.65, 25.63	0.25	2.79	−4.48, 7.96	0.18	7.40	−12.24, 20.74
Scratch body while spraying	0.21	9.87	−15.81, 28.19	−0.53	3.12	−11.47, 2.46	0.37	8.29	−7.79, 29.17
Read pesticide labels	−0.74	17.75	−88.12, 8.98	−0.17	5.62	−12.85, 12.21	−0.75	14.91	−81.47, −14.99 *
Mix with bare hands	0.16	7.80	−12.23, 22.54	0.32	2.47	−2.58, 8.43	0.07	6.55	−12.37, 16.83
Wash hands after mixing pesticides	−0.50	9.78	−38.94, 4.65	0.45	3.09	−2.38, 11.42	−0.64	8.21	−39.97, −3.35 *
Shower after spraying	−0.30	15.85	−23.60, 47.04	−0.004	5.02	−11.23, 11.14	0.30	13.31	−17.90, 41.43
Wash clothes on the same day of spraying	−0.20	6.92	−20.16, 10.69	0.03	2.06	−4.34, 4.85	0.03	5.47	−11.45, 12.94

* Statistical significance.

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
