# Peer review of "Association between Organophosphate Pesticide Exposure and Insulin Resistance in Pesticide Sprayers and Nonfarmworkers"

_ijerph, 2020, doi:10.3390/ijerph17218140_

Round 1

Reviewer 1 Report

I have several comments. Lines 96-97. How were the 36 sprayers and 42 nonsprayers selected? Why were not all 142 sprayers and 143 nonsprayers included? Please explain. Tables 4 and 5 titles mention sprayers and general population. Do you mean sprayers and nonsprayers? Why were they combined? Would it not be better to analyze them separately? It would be a good idea to have an English editor check the syntax, especially in the Discussion. For example, lines 234-235 would sound better as follows: "DAP metabolite levels in the current study were also lower than those reported in a study of pesticide sprayers in Greece (38), which could be due to the lower proportion of PPE use. " There are several more examples in the Discussion that need improved syntax.

Reviewer 2 Report

Need English language editing 

This manuscript is a resubmission of an earlier submission. The following is a list of the peer review reports and author responses from that submission.

Round 1

Reviewer 1 Report

Extensive English editing is needed.  Examples:

line 29     'both groups' replace with 'either group'

line 30     No significant association was observed between ........

line 78     replace 'which were' with 'including'.

line 77     replace 'who meet' with 'meeting'.

line 82      'participants that were initially selected and were observed ......

line 83      'afterward' replace with 'additionally'

line 85       'After at least 8 hours of fasting , each participant was interviewed and provided a blood sample to measure fasting blood glucose and insulin levels.

Line 93.    Several subjects were excluded from the enrollment group: 4 farmers that were not pesticide sprayers;  a sprayer with a small blood sample;3 sprayers and 7 nonsprayers because of high FBS levels.  The final groupto be analyzed, consisted of ...., including 36 sprayers and 42 nonsprayers to be analyzed for  ....

line 107   'PPE wearing' replace with 'use of PPE'. 

line 123 'lowering margin'  please explain

line 180  different, use difference

line 182   'which was'   replace with 'which were'

line 188    Pesticide sprayers have the higher proportion of ....

line 210   'no..., neither..., nor...'  replace with 'no..., either..., or...'

Line 233    not clear

line 269-284   not clear

Reviewer 2 Report

In this cross-sectional study, authors aimed to evaluate the association between organophosphate exposure and insulin resistance in pesticide sprayers and non-farmworkers in the Chiang Mai province in Thailand. A cross-sectional study was conducted and participants were pesticide sprayers  and subjects with “non involvement in agricultural activity” as described by authors; each group was composed of 150 participants. Fasting blood glucose and insulin level were measured  in all participants ; besides organophosphate metabolites were analyzed in urine samples only in one third of participants   in both groups. Authors assessed the factors associated with DAP metabolite in pesticide sprayers; they also assessed  factors associated with HOMA-IR and fasting blood glucose. Although the aim of the study is of interest it presents major flaws  and weakness.

  • In the material and methods section, the study design is not well described and more details must be given. for example authors must precize  the frequency of exposure of the pesticide sprayers. Authors must also assess whether the frequency of exposure was identical in all participant of the pesticides sprayer group.
  • In the material and method section authors must described whether pesticides sprayers have used other pesticides in the last months preceding the experiment.
  • DAP metabolites are certainly not the only type of pesticide metabolites that can be present in urine and that can be or not correlated with a metabolic parameter. Why authors did or did not assess other pesticide families in urine samples? How authors get rid of impact of the exposure to other pesticides used by sprayers in their epidemiological analysis?
  • The two groups of participants are not similar anthropometric and socioeconomic characteristics and this can add a bias in the results of the study; how authors get rid of this problem?
  • Authors did not precise whether they got one or multiple daily urine samples and in this last case how many days the collection of the daily urine was made?
  • Table 4 and table 5 are not clear. They show factors associated with HOMA-IR and fasting blood glucose but in which group?
  • It does not appear clearly why authors have investigated the factors associated with HOMA-IR
  • The discussion is disproportionate compared the data presented
  • Bibliographic references need to be updated

Reviewer 3 Report

Introduction

  • Statement that animal studies how evidence of chronic OP exposure reduces insulin resistance should be supported. From title of studies in reference list it appears that most studies relate to malathion and not OPs in general.

Methods

  • In order to sum DAPs, these need first to be converted to molar concentrations. This should be stated in methods
  • It is recommended to substitute undetectable DAP concentrations with half the LOD or LOD divided by square root of 2

Results

  • Table 2, what are the units for urinary concentration? Are these creatinine corrected?

Discussion

  • Consider not reporting DMP results. In light of the very high LOD these results provide inaccurate picture of low exposure. LOD for DMP in this study is about 5 times higher than reported in Health Canada (5th cycle report). For other metabolites, LOD is comparable to that reported by Health Canada.
  • Page 9, line 240, sprayers reported higher organic consumption. Is "lower" a mistake here?
  • The fact that over 80% of non-farmworkers are exposed to OPs, possibly chlorpyrifos is of concern. This is should be mentioned in the discussion, are there potential health effects? recommendations?